# Paclitaxel plus carboplatin and durvalumab with or without oleclumab for women with previously untreated locally advanced or metastatic triple-negative breast cancer: the randomized SYNERGY phase I/II trial

Chemo-immunotherapy is the first-line standard of care for patients with PD-L1 positive metastatic triple-negative breast cancer (mTNBC). SYNERGY (NCT03616886) is a dose-finding phase I and a randomized phase II, open-label trial evaluating if targeting the immunosuppressive adenosine pathway can enhance the antitumor activity of chemo-immunotherapy. The phase I part included 6 patients with untreated locally-advanced or mTNBC to determine the safety and recommended phase II dose of the anti-CD73 antibody oleclumab in combination with the anti-PD-L1 durvalumab and 12 cycles of weekly carboplatin and paclitaxel. In the phase II part, 127 women were randomized 1:1 to receive chemo-immunotherapy, with (arm A) or without (arm B) oleclumab. The primary endpoint was the clinical benefit rate at week 24, defined as stable disease, partial or complete response per RECIST v1.1. Secondary endpoints included objective response rate, duration of response, survival outcomes (progression-free survival and overall survival), and safety. The trial did not meet its primary endpoint, as the 24-week clinical benefit rate was not significantly improved by adding oleclumab (43% vs. 44%, $p = 0.61$). Exploratory median progression-free survival was 5.9 months in arm A as compared to 7.0 months in arm B ($p = 0.90$). The safety profile was manageable in both arms.

Defined by the absence of expression of endocrine receptors and lack of human epidermal growth factor receptor 2 (HER2) gene amplification, triple-negative breast cancer (TNBC) represents 15% to 20% of all breast cancers. TNBC has the worst outcome of all breast cancer subtypes, aggressive clinical behavior, a high rate of early relapses, and limited treatment options[1]. Cancer immunotherapy with monoclonal antibodies blocking the inhibitory programmed cell death-1 pathway (PD-1/PD-L1) significantly improved the survival of patients with TNBC. Although PD-1/PDL1 immune checkpoint inhibitors (ICI) evaluated as monotherapy demonstrated limited activity in early-phase trials, when combined to chemotherapy as first-line treatment in metastatic TNBC (mTNBC) in phase III trials, patient outcomes were improved[2]. Since the survival benefit was limited to patients with PD-L1 positive tumors, the United States Food & Drug Administration (US FDA) and the European Medicines Agency (EMA) approved chemo-immunotherapy combination in this specific indication[3,4]. Of note, in the neoadjuvant setting, pathological complete response rates at surgery were increased with the addition of PD-1/PD-L1 ICI to standard chemotherapy, irrespective of tumor PD-L1 status. This benefit resulted in an improvement of the event-free survival in the KEYNOTE-522 phase III

✉ e-mail: laurence.buisseret@bordet.be

trial and the combination was recently regulatory-approved in this setting[5]. However, despite these remarkable advances, not all patients derive the same benefit from ICI, and several research efforts are ongoing to identify new strategies to enhance the activity of ICI.

Furthermore, a subset of patients treated with ICI in the early setting (~16% at 3 years) will relapse, and new treatment strategies are needed[5]. In melanoma, the combined inhibition of PD-1 and CTLA-4 or LAG3 highlighted the clinical potential of combining immunotherapeutic agents with synergistic mechanisms of action[6,7]. In TNBC, preclinical evidence suggests that targeting the immunosuppressive adenosinergic pathway could enhance the efficacy of PD-1/PD-L1 and CTLA-4 ICI[8]. In tumors, this pathway is induced by tissue hypoxia, chronic inflammation, and oncogenic pathways. It is responsible for the conversion of pro-inflammatory extracellular adenosine triphosphate (ATP) released by cancer cells into extracellular immunosuppressive adenosine through the concerted action of the cell-surface ectonucleotidases CD39 and CD73[9]. Extracellular adenosine activates adenosine receptors on immune cells, impairing their anti-tumor activity and tumor cells, promoting their survival and metastatic properties[9]. CD73 is expressed on the surface of tumor cells, stromal cells, and immune cells in the tumor microenvironment (TME). It is associated with a worse outcome in several solid cancers, including TNBC, as demonstrated in large phase III adjuvant clinical trial[10]. Based on the prognostic impact of CD73 in TNBC and preclinical proof-of-concept studies, targeting CD73 in order to relieve adenosine-mediated immunosuppression seems a promising strategy in TNBC[11]. Oleclumab (MEDI9447) is a first-in-class human monoclonal antibody IgG1λ inhibiting CD73 and blocking the conversion of extracellular ATP into adenosine, thus decreasing the immunosuppression in the TME. In phase I and II trials, the combination of oleclumab with durvalumab showed antitumor activity and a manageable safety profile in patients with solid tumors[12,13]. In a randomized phase II trial in patients with unresectable stage III non-small cell lung cancer (NSCLC) and no progression after concurrent chemo-radiotherapy, combining oleclumab with durvalumab (a Fc optimized IgG1κ monoclonal antibody directed against PD-L1) increased objective response rate and 12-month progression-free survival (PFS) compared to durvalumab alone[12].

The randomized phase I/II SYNERGY trial (NCT03616886) evaluated the safety and efficacy of adding oleclumab to the combination of durvalumab with chemotherapy in patients with untreated locally-advanced or mTNBC[14]. The phase I part determined the recommended phase II dose (RP2D) of oleclumab in combination with durvalumab and intravenous chemotherapy consisting of 12 weeks of paclitaxel (80 mg/m²) and carboplatin (AUC 2) followed by maintenance with dual immunotherapy. The phase II part of the trial randomized patients to the 12 weeks of chemotherapy in combination with durvalumab with (arm A) or without oleclumab (arm B) (Supplementary Fig. 1). Patients with either PD-L1 positive or negative and either CD73 positive or negative tumors assessed centrally by immunohistochemistry (IHC) on a baseline tumor sample were eligible to enter the trial. The primary endpoint was the clinical benefit (CB) by investigator assessment per RECIST v1.1 at week 24 with the hypothesis that adding oleclumab in arm A would increase the CB rate (CBR) from 40 to 60%. The secondary study endpoints were objective response rate (ORR), duration of response (DOR), survival outcomes, and evaluation of the safety of the treatment combination. Here we report the results of the phase I and II of the SYNERGY clinical trial.

## Results
### Patient characteristics
The study recruited women with previously untreated, inoperable, locally advanced, or mTNBC who were 18 years or older, with an Eastern Cooperative Oncology Group (ECOG) Performance Status of 0 or 1, and at least one measurable lesion per RECIST v1.1. Patients agreed to provide tumor tissue for central assessment of PD-L1 and CD73 IHC

status prior to randomization for stratification and a second biopsy at week 3 for translational research purposes. For patients with recurrent TNBC, a disease-free interval of at least 6 months was required. Main exclusion criteria included untreated brain metastases and/or carcinomatous meningitis, and medical contra-indications to anti-PD-L1 ICI such as ongoing steroids therapy of more than 10 mg/day of prednisone or its equivalent. A full list of inclusion and exclusion criteria is presented in the "Methods" section.

### Phase I part results
Between January 2019 and February 2019, six patients were included in the phase I part of the study. All enrolled patients had recurrent TNBC (Supplementary Table 1). No Dose-limiting toxicities were observed during the phase I part, and the RP2D of oleclumab in combination with durvalumab and chemotherapy with paclitaxel and carboplatin was 3000 mg[15]. However, 5 out of 6 enrolled patients experienced grade 3-4 neutropenia, leading to a carboplatin dose reduction in phase II from AUC 2 to AUC 1.5. Regarding the tumor response, four patients presented a non-progressive disease a week 24. Moreover, two are still under maintenance treatment more than four years after the study entry.

### Phase II part results
Between June 2019 and June 2021, a total of 129 patients from 16 centers in Belgium and France were randomized and treated with 12 weekly administrations of paclitaxel 80 mg/m² with carboplatin AUC 1.5, combined with durvalumab 1500 mg every 4 weeks with (arm A) or without (arm B) oleclumab 3000 mg every 2 weeks for five administrations then every four weeks (Supplementary Fig. 1). Figure 1 shows the CONSORT flow diagram with 127 evaluable patients for the primary endpoint: 63 in arm A and 64 in arm B. The median age was 58 years (interquartile range (IQR) 47.0–67.0) in arm A and 55 years (IQR 47.0–67.0) in arm B. There was no significant imbalance between the two arms. However, de novo disease presentation (17 in arm A (27%) vs. 24 patients in arm B (37.5%), p = 0.26) and prior carboplatin exposure (4 in arm A (8.7%) vs. 10 in arm B (25%), p = 0.08) were numerically more frequent in arm B. Around one-third of patients in each arm had liver metastases. At randomization, PD-L1 IHC status assessed using the VENTANA SP263 assay was positive in 58.7% of patients in arm A and 64.1% in arm B, whereas CD73 IHC status was positive in 27.0% in arm A and 34.4% in arm B. The double positivity for both biomarkers was 22.2% and 21.9% in arms A and B, respectively. Table 1 summarizes the baseline clinicopathological characteristics of the evaluable patients.

### Efficacy assessment
At the data cut-off of January 31, 2023 and after a median follow-up time of 16.5 months (IQR 8.5-21.3), the CBR at week 24 was 43% in arm A and 44% in arm B (p = 0.61 [one-sided p-value for Fisher's exact test], Fig. 2a). The distribution per type of response according to RECIST v.1.1 at week 24 is shown in Fig. 2b. Four CR were observed in arm A and one in arm B. The objective response rate (ORR) was 63.5% in arm A and 64.1% in arm B (one-sided p = 0.6). This negative result for the primary endpoint was expected as the prespecified boundary for futility was crossed at the interim analysis performed when 68 patients were evaluable for the primary endpoint at week 24 with a CBR of 48% in arm A and 51% in arm B (z-score = −0.2574 which is below the stopping boundary of 0.074); p value = 0.69 (one-sided Fisher's exact test)[16]. Data were reviewed by an independent data monitoring committee (IDMC), and further recruitment in both arms was stopped on June 22, 2021, according to their recommendation. As the decision to halt recruitment was not related to safety, patients still receiving the study treatment could continue either with oleclumab (arm A) or without after discussion with their treating physician. All patients decided to continue the allocated study treatment except three patients in arm A who chose to stop oleclumab and continue durvalumab alone after

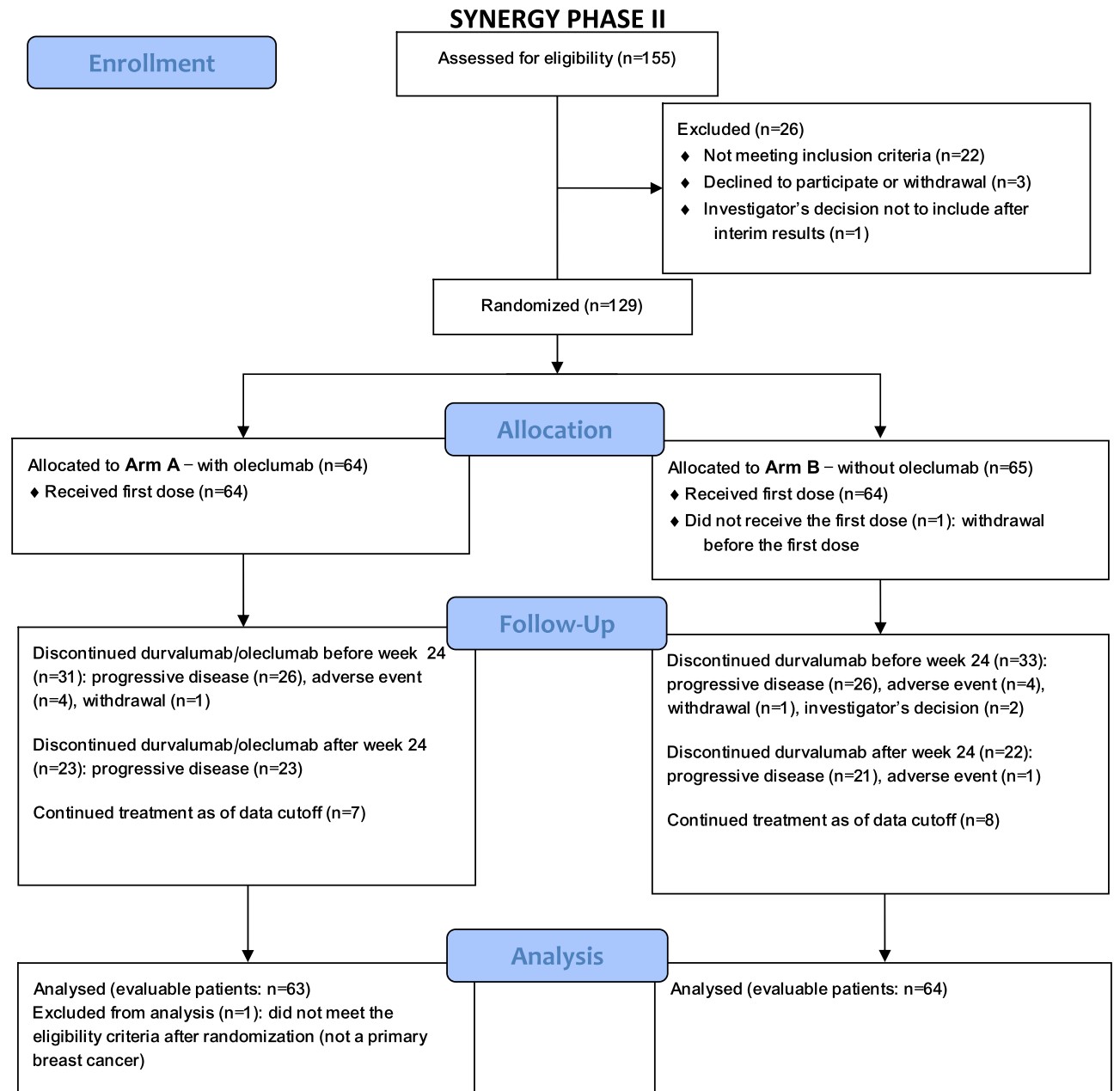

**Fig. 1 | Patient flow diagram.** Flowchart of patients' progress though the phases of the trial.

having received combined immunotherapy maintenance for more than one year. All analyses outside the primary endpoint are exploratory, as performed on the 127 evaluable patients out of 136 initially required by the sample size estimation for the primary endpoint assessment.

The prespecified biomarker analyses comprised PD-L1 and CD73 IHC status prospectively assessed prior randomization on the most recent tumor tissue available (baseline metastatic ($n = 82$), baseline primary ($n = 30$), or archived primary tumor ($n = 15$) in case the baseline tissue was not evaluable) (Supplementary Table 2). No statistical differences between arms in CBR were observed according to these biomarkers (Fig. 2c, d). Exploratory unplanned analyses were also performed in subgroups. A higher CBR regardless of treatment arm was recorded in patients with de novo metastatic disease, normal baseline LDH level and lower neutrophils to lymphocytes ratio (Supplementary Fig. 2).

Median PFS and OS were not significantly different between treatment arms, with a median PFS numerically shorter in arm A (5.9 (95% CI: 5.2–8.0) vs. 7.0 (95% CI: 5.4–10.0) months in arm B, $p = 0.90$; one-sided log-rank test) and median OS numerically longer in arm A (25.1 (95% CI: 16.6–NR) vs. 19.3 (95% CI: 16.9–NR) months in arm B, $p = 0.62$; one-sided log-rank test). Survival data are shown in Fig. 2e, f. In exploratory subgroup analyses, PFS was longer in the overall PD-L1 positive compared to PD-L1 negative population (median PFS of 7.1 vs. 5.4 months, $p = 0.05$; two-sided log-rank test) and this increased PFS for patients with PD-L1 positive tumors appeared in arm A (Supplementary Figs. 3a and 4). The interaction test between treatment arms and PD-L1 status was not significant ($p = 0.07$; Cox regression model). There were no differences in PFS or OS between arms in the PD-L1 positive subgroup (Supplementary Fig. 3c, d). No survival differences were observed according to CD73 status (Supplementary Fig. 5a–d).

## Table 1 | Baseline patient characteristics

| Characteristics | Arm A With oleclu-mab (N = 63) | Arm B Without oleclu-mab (N = 64) | p value |
|---|---|---|---|
| Age, in years | | | 0.81 |
| Median (IQR) | 58.0 (47.0, 67.0) | 55.0 (47.0, 67.0) | |
| <40 | 7 (11.1%) | 7 (10.9%) | |
| 40–65 | 36 (57.1%) | 40 (62.5%) | |
| >65 | 20 (31.7%) | 17 (26.6%) | |
| ECOG performance status | | | 0.45 |
| 0 | 39 (62.9%) | 44 (68.8%) | |
| 1 | 23 (37.1%) | 19 (29.7%) | |
| 2 | 0 (0.0%) | 1 (1.6%) | |
| Missing | 1 (1.6%) | 0 (0.0%) | |
| Disease presentation | | | 0.26 |
| De novo metastatic | 17 (27.0%) | 24 (37.5%) | |
| Recurrent metastatic | 46 (73.0%) | 40 (62.5%) | |
| Disease-free interval (% of recurrent) | | | NT |
| <6 months | 0 (0%) | 1 (2.5%) | |
| 6–12 months | 3 (6.5) | 0 (.0%) | |
| >12 months | 43 (93.5%) | 39 (97.5%) | |
| Prior chemotherapy (early-setting, % of recurrent) | | | NT |
| Prior anthracycline | 39 (84.8%) | 37 (92.5%) | |
| Prior taxane | 36 (78.3%) | 33 (82.5%) | |
| Prior carboplatin | 4 (8.7%) | 10 (25.0%) | |
| Metastatic sites (number) | | | 0.67 |
| <2 | 14 (22.2%) | 12 (18.8%) | |
| ≥2 | 49 (77.8%) | 52 (81.3%) | |
| Metastatic sites (targets) | | | |
| Liver | 20 (31.7%) | 20 (31.3%) | 1 |
| Bone | 17 (27.0%) | 12 (18.8%) | 0.3 |
| Lung | 31 (49.2%) | 26 (40.6%) | 0.37 |
| Lymph nodes | 34 (54.0%) | 39 (60.9%) | 0.48 |
| Others (skin, perito-neum, ...) | 29 (46%) | 33 (51.6%) | 0.6 |
| Germline BRCA mutational status | | | 0.81 |
| BRCA1/2 mutation | 7 (11.1%) | 4 (6.3%) | |
| Absence of mutation | 30 (47.6%) | 34 (53.1%) | |
| Unknown | 26 (41.2%) | 26 (40.6%) | |
| LDH | | | 0.48 |
| ≤UNL | 35 (55.6%) | 31 (48.4%) | |
| >UNL | 13 (20.6%) | 12 (18.8%) | |
| NLR | | | 0.82 |
| ≤5 | 51 (81%) | 53 (82.8%) | |
| >5 | 12 (19%) | 11 (17.2%) | |
| PD-L1 status | | | 0.59 |
| Negative | 26 (41.3%) | 23 (35.9%) | |
| Positive | 37 (58.7%) | 41 (64.1%) | |
| Positivity by sample type | | | |
| Metastasis | 27/43 (62.8%) | 23/39 (59%) | |
| Primary (de novo) | 7/13 (53.8%) | 12/17 (70.6%) | |
| Primary (archived) | 3/7 (42.9%) | 6/8 (75%) | |
| CD73 status | | | 0.44 |
| Negative | 46 (73%) | 42 (65.6%) | |
| Positive | 17 (27.0%) | 22 (34.4%) | |

## Table 1 (continued) | Baseline patient characteristics

| Characteristics | Arm A With oleclu-mab (N = 63) | Arm B Without oleclu-mab (N = 64) | p value |
|---|---|---|---|
| Positivity by sample type | | | |
| Metastasis | 14/43 (32.5%) | 14/39 (35.9%) | |
| Primary (de novo) | 2/13 (15.4%) | 3/17 (17.6%) | |
| Primary (archived) | 1/7 (14.3%) | 5/8 (62.5%) | |
| Combinations | | | 0.24 |
| PD-L1 positive and CD73 positive | 14 (22.2%) | 14 (21.9%) | |
| PD-L1 negative and CD73 positive | 3 (4.8%) | 8 (12.5%) | |
| PD-L1 positive and CD73 negative | 23 (36.5%) | 27 (42.2%) | |
| PD-L1 negative and CD73 negative | 23 (36.5%) | 15 (23.4%) | |

Note: PD-L1 and CD73 IHC status were defined positives if ≥1% of stained tumor and/or immune cells for PD-L1 (clone SP263) and of tumor and/or stromal cells for CD73 (clone EPR6115) of any intensity relative to the tumor and stroma area were identified. All tests are two-sided. No type-1 error adjustments for multiple comparisons. Wilcoxon non-parametric test is used to analyze continuous variables. Fisher's exact test is used to analyze categorical variables.
*IQR* interquartile range, *ECOG* Eastern Cooperative Oncology Group, *LDH* lactate dehydrogenase, *NLR* neutrophil-to-lymphocyte ratio, *PD-L1* programmed cell death ligand 1.

The median duration of response was 5.6 months (IQR 3.7–9.6 months) in arm A and 5.9 months (IQR 3.7–12.0 months) in arm B. A subset of 28 patients (10 in arm A (15.9%) and 18 in arm B (28.1%)), were considered as long-term responders with no disease progression or death 12 months after randomization. Six of these 28 long responders had achieved a PFS ≥2 years at the time of data cut-off. The duration of treatment and responses of these 28 patients are represented in Fig. 3. Of note, three long responders who experienced PD and were clinically stable were allowed to continue therapy beyond progression. At the data cut-off, 15 of these 28 patients (7 in arm A and 8 in arm B), including the three patients who continued therapy beyond progression, are still under immunotherapy maintenance. Clinical factors that influenced the duration of response of these 28 patients were disease presentation with de novo metastatic disease (18/28), which was associated with long-term benefit. In contrast, younger age (<40 years) seems to be correlated with shorter response duration. Of note, 67.9% of the long responders were in PR, 3.6% in CR, and 28.6% in SD at the first tumor evaluation at week 8. Resection of the primary tumors for patients presenting de novo metastatic disease was allowed per protocol after week 24 evaluation. The table with baseline characteristics of the 28 long responders compared with patients included in the trial is shown in Supplementary Table 3.

### Safety assessment and treatment exposure

The most relevant adverse events (AEs) are shown in Fig. 4. The prevalence of AEs, mostly hematological, was higher when chemotherapy was administered with immunotherapy (i.e., first 12 weeks). A numerically higher proportion of neutropenia, without statistical significance, was observed in arm A with oleclumab (52.4% vs. 42.2% grade 3–4, $p = 0.29$) both in patients with de novo (47.1% in arm A vs. 36.0% in arm B) and chemotherapy pretreated recurrent disease (54.3% in arm A vs. 45.0% in arm B). The AEs of special interest (AESI), defined as immune-mediated as specified in the protocol, were monitored. In arm A, AESI included events as thrombotic events related to the cardiovascular effects of oleclumab. Grade 3 AESI occurred in 15.9% in arm A vs. 12.5% in arm B, and their prevalence decreased to 6.4% vs. 0% during immunotherapy maintenance. Supplementary Table 4 describes grade

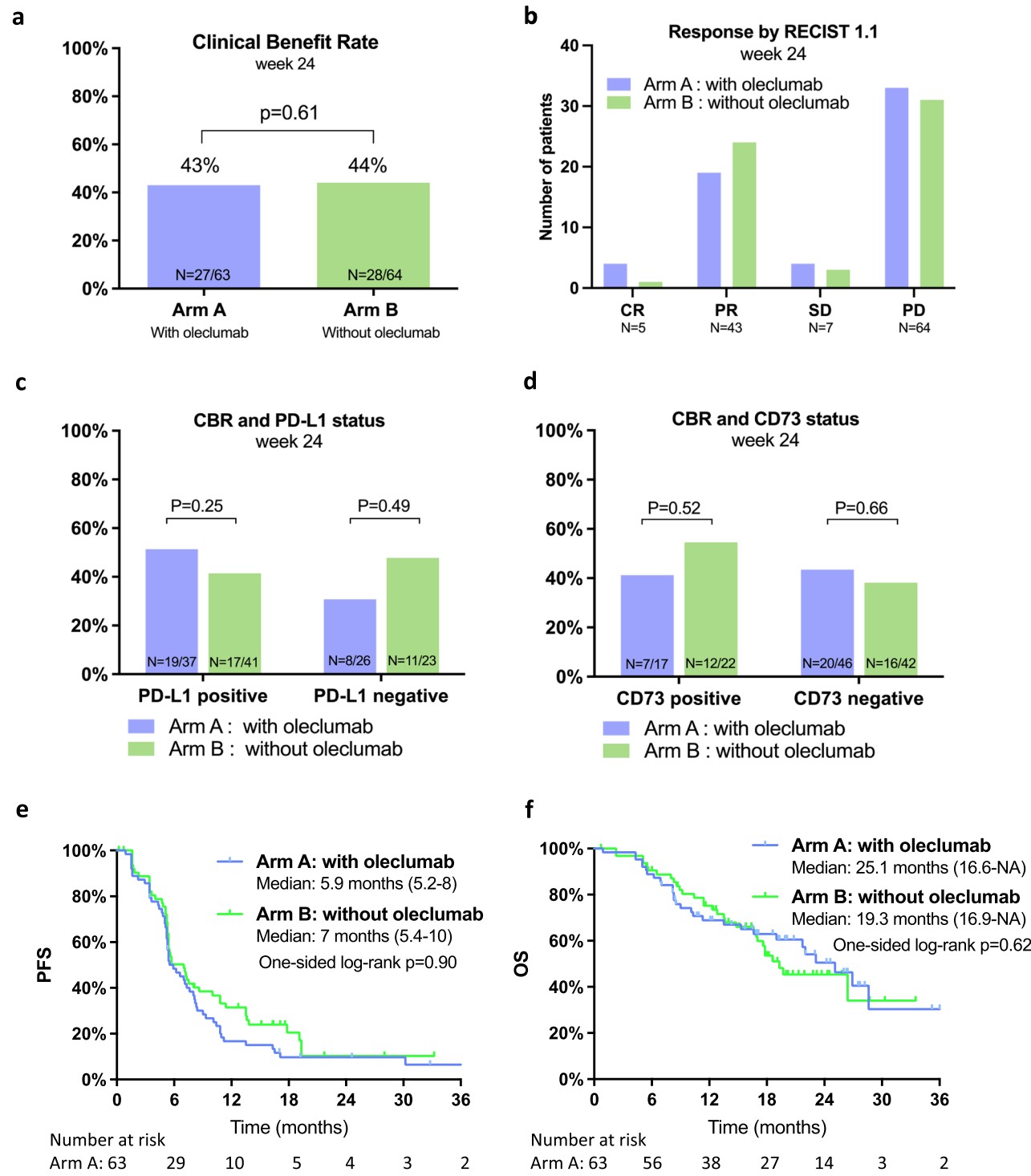

**Fig. 2 | Efficacy endpoints. a** Primary endpoint result: clinical benefit rate (CBR), including patients in stable disease (SD), partial response (PR) and complete response (CR) at week 24 per treatment arm. Statistical significance was tested with a one-sided Fisher's exact test. **b** Distribution of responses as defined by RECIST 1.1 among patients evaluable at week 24. **c** CBR according to PD-L1 status. Two-sided

Fisher's exact test. **d** CBR according to CD73 status. Two-sided Fisher's exact test. **e** Kaplan–Meier estimates of progression-free survival (PFS) by RECIST v1.1 according to treatment arm. One-sided log-rank test. **f** Kaplan–Meier estimates of overall survival (OS) according to treatment arm. One-sided log-rank test. Source data are provided as a Source data file.

3 AESI. There were no grade 4 AESI in both arms. Serious AEs occurred in 33.3% in arm A and 29.7% of arm B. No grade 5 toxicity was observed in this study. Four patients in arm A (6.3%) and 5 in arm B (7.8%) discontinued the study treatment due to an AE. Three of these patients were followed for efficacy and did not present with disease progression despite treatment discontinuation for more than 10 months

(Fig. 3). Oleclumab was interrupted for 2 of the 3 patients in arm A who chose to stop the study drug and continued with durvalumab alone after the interim analysis because one patient presented a grade 3 fatigue and the second patient a cortisol insufficiency (Fig. 3). Median treatment exposure was 5.4 months (IQR 2.8–9.2 months) in arm A and 4.8 months (IQR 3.0–11.8 months) in arm B.

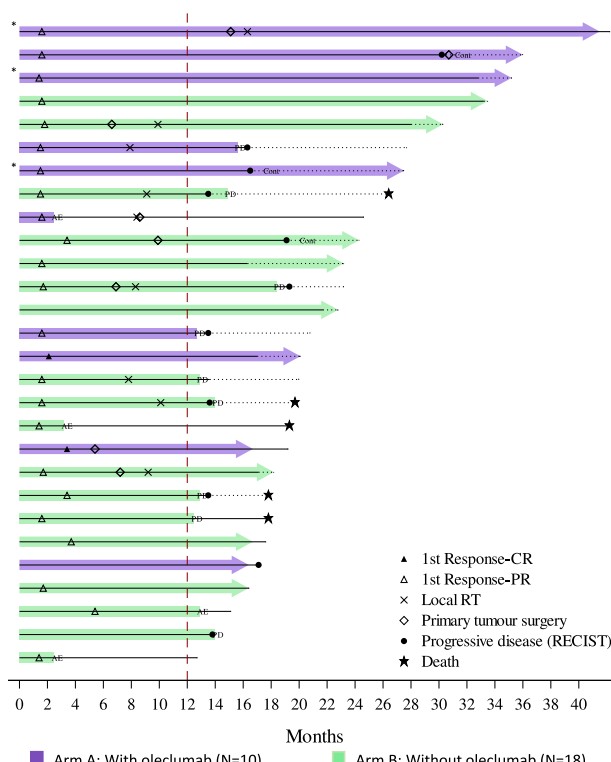

**Fig. 3 | Swimmer plot of long responders.** Long responders are defined as patients without progressive disease or remaining under study treatment beyond 12 months after study treatment first administration. Each bar represents treatment duration of a patient. Solid line represents the follow-up up to last tumor evaluation, dotted line−up to last follow-up visit or death. CR complete response, PR partial response, RT radiotherapy, PD progressive disease, AE adverse event, Cont continuation beyond PD. *Patients crossed over arm B (stop oleclumab) after protocol amendment based on the IDMC recommendations.

## Exploratory biomarker analyses

As defined in the protocol, baseline tumor biopsies (metastases; $n = 82$ and primary tumors; $n = 30$) and on tumor biopsies at week 3 (metastases; $n = 80$ and primary tumors; $n = 24$) were collected to identify potential predictive biomarkers of response (Fig. 5a). A flow diagram with the collected samples evaluable for this exploratory biomarker analysis is presented in Supplementary Fig. 6.

Baseline stromal tumor-infiltrating lymphocytes (str-TILs) levels >10% assessed on H&E stained slides were identified in 19.8% of tumor samples and were associated with a better PFS and OS with no interaction by the treatment arm (Supplementary Table 5, Fig. 5b, c, and Supplementary Fig. 7a−d). PD-L1 expression was evaluated on immune and tumor cells, and using the combined positive score (CPS). PD-L1 positive immune cells and positive tumor cells were identified in 69.4% and 7.2% of baseline samples, respectively (Supplementary Table 5). Using the CPS, 74.7% of these samples were classified as CPS > 0, 32.4% ≥ 1 and 17.1% as CPS≥10. There was a higher proportion of CPS-positive cases in arm B (without oleclumab), causing a significant imbalance between the arms ($p = 0.008$; Supplementary Table 5). Tumor tissues were collected from various organs and str-TIL levels, and PD-L1 expression were heterogenous according to metastatic sites (Supplementary Table 6). Baseline str-TILs were positively correlated with PD-L1 CPS (Spearman Rho: 0.496, $p < 0.0001$; Supplementary Fig. 3f). Interestingly, the CBR at week 24 was increased in patients with CPS ≥1 tumor (58.3% vs. 38.1%, two-sided $p = 0.066$, Supplementary Fig. 8a), and although this benefit was not significant, a longer PFS was observed in this subgroup (median PFS 13.5 vs. 5.4 months in PD-L1 CPS ≥ 1 and <1, respectively, $p = 0.0004$; two-sided log-rank test, Fig. 5d). This longer PFS in the PD-L1 positive subgroup appears in both arms (Supplementary Fig. 9). The difference in OS was not statistically significant (Fig. 5e). Of note, arm A and B had no significant survival differences in the CPS ≥ 1 and <1 subgroups (Supplementary Fig. 8c−f). Multivariate analysis, including treatment arm, str-TILs, PD-L1 CPS confirmed that disease presentation (de novo vs. recurrent) significantly influenced the CBR and the PFS (Supplementary Fig. 10).

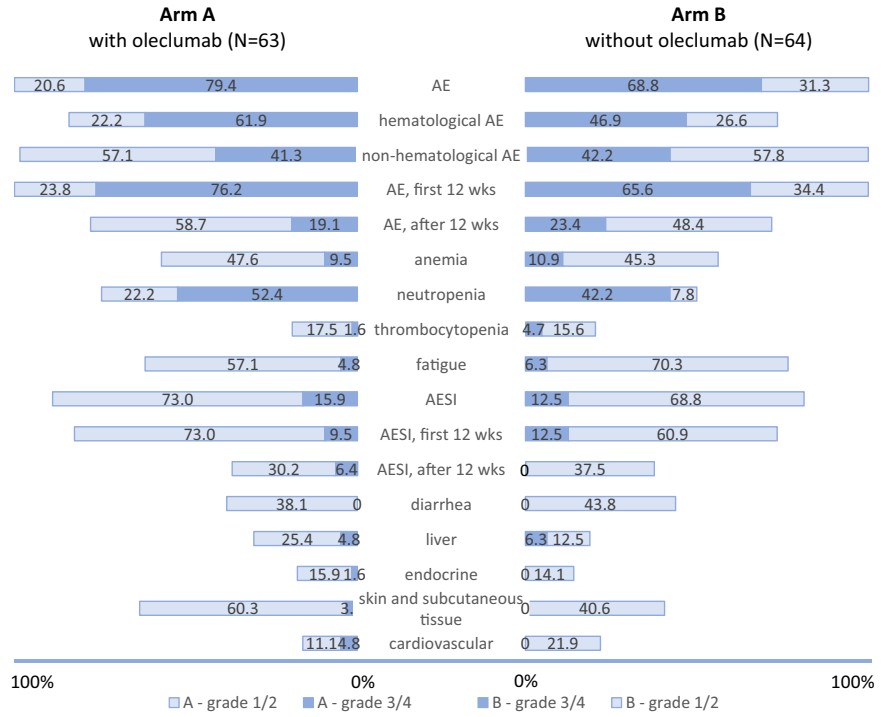

**Fig. 4 | Summary of adverse events.** Adverse events per treatment arm and per grade according to CTCAE version 5.0. AE adverse events, wks weeks, AESI adverse event of special interest.

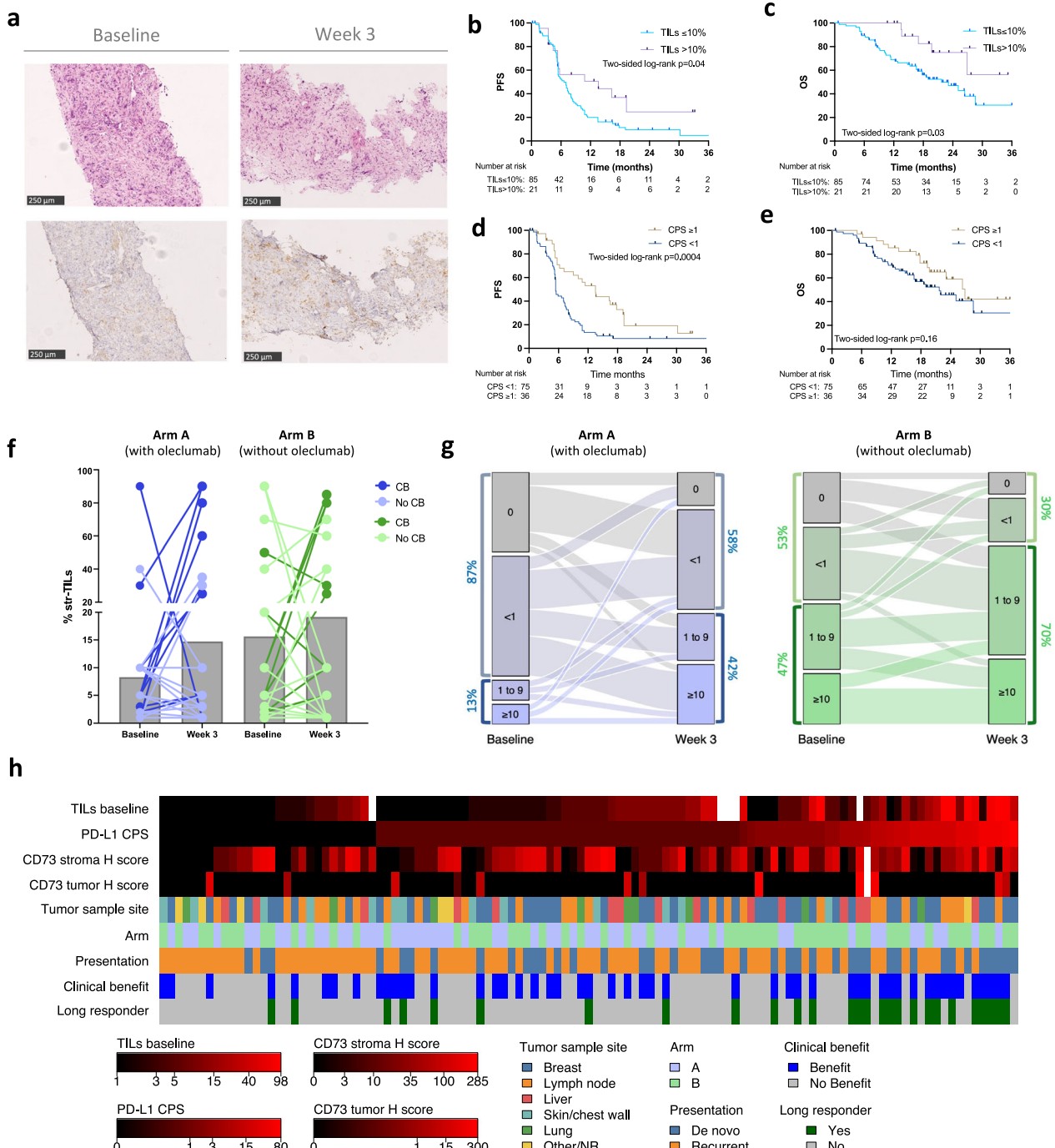

**Fig. 5 | Exploratory biomarker analyses. a** Representative H&E (upper panels) and PD-L1 (lower panels) staining of a liver biopsy (baseline, on the left, and at week 3, on the right). **b** Kaplan–Meier estimates of progression-free survival (PFS) with low (≤10%) and high (>10%) stromal tumor-infiltrating lymphocytes (str-TILs) in the overall trial population (arm A & B). Two-sided log-rank test. **c** Kaplan–Meier estimates of overall survival (OS) with low (≤10%) and high (>10%) stromal TILs. Two-sided log-rank test. **d** Kaplan–Meier estimates of PFS according to PD-L1 status using the CPS score. Two-sided log-rank test. **e** Kaplan–Meier estimates of OS according to PD-L1 status using the CPS score. Two-sided log-rank test. **f** Dynamic of str-TILs between baseline and week 3 biopsies in arm A (with oleclumab) and in arm B (without oleclumab) (35 paired samples in arm A and 30 in arm B). Patients with clinical benefit (CB) at week 24 are represented in dark blue (arm A) and dark green (arm B). **g** Alluvial plots showing the transition of CPS between paired baseline and week 3 in arm A and in arm B. **h** Oncoplot including exploratory biomarker analyses (baseline stromal str-TILs, PD-L1 CPS, CD73 tumor and stroma histological scores) and clinical variables. Source data are provided as a Source data file.

To further investigate treatment impact on str-TILs levels and PD-L1 expression dynamics, these biomarkers were retrospectively assessed on a second biopsy performed after 3 weeks of treatment (Supplementary Table 7). A str-TIL levels increase of at least 5% on the week 3 biopsy was observed in 26% of the cases (6 cases in arm A (20%) and 9 cases in arm B (32.1%)). This increase was not statistically significant ($p = 0.3969$, Wilcoxon paired test). Interestingly, a higher proportion of patients with increased str-TILs on the week 3 biopsies was observed among patients with CB at week 24 compared to patients with no benefit (35.7% vs. 6.3% in arm A, 35.7% vs. 23.5% in arm B) (Fig. 5f). A decrease in str-TILs of at least 5% was observed in 20.7% of the cases (6 cases in arm A (20%) and 6 cases in arm B (21.4%)). We

observed a shift to an upper PD-L1 CPS category on the week 3 biopsies in 55% of the cases (19 cases in arm A (61.3%) and 14 cases in arm B (48.3%)), in patients with CB and without CB at week 24 with no difference between the two arms (Fig. 5g and Supplementary Fig. 11). The shift in CPS on week 3 biopsies was mainly related to an increase in the proportion of PD-L1 positive tumor cells compared to baseline samples (Supplementary Table 5).

CD73 IHC expression was mostly observed on stroma cells, with only a few cases with positive tumor cells ($n = 9/82$ metastases, $n = 3/30$ primary tumors) (Supplementary Fig. 12). A semi-quantitative assessment of CD73 expression on stromal cells (Histological score (H-score): % of positive stromal cells × staining intensity) showed a higher expression of CD73 in arm B (without oleclumab) mainly driven by CD73 expression in the stroma of primary tumors from patients with de novo metastatic disease (Supplementary Table 8). There was no correlation between str-TILs and CD73 stromal expression in our cohort. CD73 did not influence the response to the study treatment as CD73 stroma H-scores were similar in patients with CB at week 24 and those with no benefit in both arms (Supplementary Fig. 12c–f). A summary oncoplot of exploratory biomarkers and clinical variables is shown in Fig. 5h.

## Discussion

The combination of chemo-immunotherapy with PD-1/PD-L1 ICI became the standard of care for patients with PD-L1 positive advanced or mTNBC in 2019 following results from IMpassion130 and KEYNOTE-355 phase III trials[4,17]. New therapeutic strategies are needed to expand the patient population who might benefit from immunotherapy and to further enhance the efficacy of currently approved immunotherapies. The SYNERGY trial is a randomized study investigating the addition of a second ICI, an adenosine pathway inhibitor (oleclumab), to increase the benefit of chemo-immunotherapy in advanced TNBC. In this trial, adding oleclumab to chemo-immunotherapy for unselected patients with untreated advanced TNBC did not increase the CBR at week 24. All patients with TNBC, regardless of PD-L1 expression, were eligible as results from phase III randomized trials testing immunotherapy in this setting had not been reported when the trial was designed in 2017 and also to investigate if oleclumab might overcome immunosuppression in PD-L1 negative tumors. Consequently, our study including only 78 patients with PD-L1 positive tumors cannot answer whether the addition of oleclumab improves outcome in a population who today would be eligible for first-line chemo-immunotherapy (i.e., PD-L1-positive tumors). In NSCLC, the combination of oleclumab and durvalumab was evaluated in the neo-adjuvant setting, and responses were associated with baseline tumor PD-L1 and CD73 expression[18]. In unresectable stage III NSCLC, the combination tested in the COAST phase II trial enhanced the anti-tumor response and improved PFS independently of PD-L1 status, leading to a phase III study initiation in this indication[12,19].

In advanced TNBC, the combination of oleclumab plus durvalumab was evaluated in the BEGONIA trial (NCT03742102), a multi-cohort basket study testing various treatment combinations, including paclitaxel, durvalumab, and oleclumab. In the BEGONIA trial, the confirmed ORR was 45.5%, irrespective of PD-L1 status[20]. In the SYNERGY trial, the CBR was also independent of PD-L1 status, but a longer PFS was observed in the overall PD-L1 positive compared to the PD-L1 negative population consistent with data from landmark phase III chemo-immunotherapy trials. In our study, PD-L1 IHC status was prospectively defined with the VENTANA SP263 assay, and 61.4% of the cases were defined positive in the overall study population with a threshold of ≥1% of any positive cells relative to the tumor and stroma area. This threshold and scoring system might have underestimated the prevalence of PD-L1 positivity. Indeed, in an analytical study comparing three PD-L1 assays in TNBC tumor samples from the phase III IMpassion130 clinical trial, PD-L1 positivity with the clone SP263 was

74.9% using a threshold ≥1% of the tumor area and peri-tumoral stroma only occupied by PD-L1-positive immune cells[21]. In our post hoc exploratory biomarker analyses, PD-L1 prevalence on immune cells was similar, with 70.6% being positive. Interestingly, the re-assessment of PD-L1 according to the CPS identified a subgroup of patients with a higher PFS benefit and demonstrate that despite the prospective PD-L1 assessment prior randomization, treatment arms were imbalanced for the CPS. These findings confirm that PD-L1 analysis is challenging and that identifying the optimal PD-L1 assay and scoring system would be of clinical relevance[22,23]. Moreover, EMA does not link a drug to one specific IHC assay, and there is certainly a significant variability depending on the antibody used in the selection of patients for ICI in routine clinical practice.

Also noteworthy is the fact that, as in the ALICE trial evaluating atezolizumab plus anthracycline-based chemotherapy in mTNBC, long-lasting responses were observed in patients with PD-L1 negative disease[24]. We identified 28 long-responders with a PFS≥12 months. Among them, four patients had a PD-L1 negative tumor (14.3%), and 10 (35.7%) had a CPS < 1. Of note, 15 patients were still under treatment at the data cut-off. One of the challenges for these patients with long responses is determining the optimal duration of maintenance immunotherapy. Currently, there are no specific guidelines for the duration of immunotherapy in metastatic breast cancer, and ICI are continued as long as they benefit, providing they don't have excessive toxicity[25].

The safety profile was consistent across the two arms and with data reported from other trials evaluating chemo-immunotherapy in advanced TNBC with similar rates of treatment discontinuation because of AEs. However, we noticed a slight increase in neutropenia in arm A, both in patients with de novo or recurrent disease without compromising chemotherapy exposure. Adenosine signaling and extracellular ATP are involved in neutrophil chemotaxis, and activation of adenosine receptors on neutrophils delays their apoptosis[26,27]. Thus, the blockade of ATP degradation into adenosine might interfere with neutrophil regulation and impair the recovery of chemotherapy-induced neutropenia. Of note, neutropenia was not associated with oleclumab without concurrent chemotherapy[12].

Although both arms were reasonably well balanced for baseline characteristics, there were a few imbalances, such as a lower proportion of patients with de novo mTNBC in arm A and a higher proportion of patients with recurrent disease previously exposed to carboplatin in arm B. Additional limitations of this study include that it was open-label, responses were investigator-assessed (rather than central-reviewed), and the study was not powered to assess a survival benefit. As a consequence of the interim analysis and the stop of accrual, the number of evaluable patients was limited. Therefore, all analyses outside of the primary endpoint are exploratory. As a result of the negative interim analysis, three patients from arm A, who had undergone maintenance immunotherapy for over a year, chose to stop taking oleclumab. The longer follow-up provides more data on the survival benefit of study treatments. Indeed, as observed with ICI use in routine, survival outcomes might be significantly improved despite a modest volumetric anti-tumoral response[28]. In landmark phase III trials, chemotherapy was continued until progression or toxicity and consisted of nab-paclitaxel in IMpassion 130, paclitaxel in IMpassion 131, and nab-paclitaxel, paclitaxel or gemcitabine-carboplatin according to physician's choice in the KEYNOTE-355 with more than 50% of the patients treated with the doublet[17,29,30]. In our trial, the chemotherapy backbone with weekly carboplatin and paclitaxel administrated for a maximum of 12 weeks aimed to relieve disease symptoms, induce a rapid tumor shrinkage, and prime an immune response through immunogenic cell death[31]. This design allowed for a short exposure to cytotoxic treatment without compromising response rate or survival, which were comparable to those reported in the KEYNOTE-355 and IMpassion130 trials[32]. The strategy of ICI maintenance was also

explored in the SAFIR02 trial with significant improvement of OS among *CD274* overexpressing population, despite the absence of difference based on the PD-L1 status[33]. Further studies are warranted to better define the optimal choice and duration of chemotherapy backbone when combined with immunotherapy as well as the optimal parameters to evaluate efficacy and long-term benefit.

In our previous work, we showed that the expression of the adenosine-producing enzymes (CD39, CD73) was mainly observed in the mesenchymal stem-like subtype representing 15% of early TNBC and was associated with an immune margin-restricted subtype[34]. CD73 IHC expression was evaluated on baseline samples in the SYNERGY study to identify its predictive value. We observed a low frequency of cases with CD73-positive tumor cells, potentially due to a low sensitivity of the clone EPR6115 to detect CD73-positive epithelial cells. CD73 on stromal cells, specifically in cancer-associated fibroblasts, has also been associated with immunosuppression and immunotherapy resistance[35]. In our cohort, stromal CD73 expression was higher on primary breast tumors compared to metastases but was not statistically associated with clinical outcomes. It was also expected that oleclumab might increase immune infiltration in post-treatment biopsies. Our biomarker analysis showed a global increase in TILs at week 3 in both arms that was more frequently observed in patients with clinical benefit, whereas the upregulation of PD-L1 expression was not associated with improved responses. In an ongoing analysis by single-cell RNA sequencing on tumors from a subset of patients from the SYNERGY trial, Metoikidou C. and colleagues observed distinct transcriptomic and TCR-clonal patterns in responders vs. non-responder patients and between treatment arms (personal communication). These analyses will shed light on the immune parameters associated with outcome to chemo-immunotherapy. The biological material collected during the SYNERGY trial will allow further translational analyses that will help to understand the characteristics associated with response to the study treatment and are warranted to better define which patients may derive benefit from immunomodulation of the adenosine pathway. Of note, several drugs targeting the adenosine-generating enzymes (CD39 and CD73) or the adenosine receptors have been developed and are currently under clinical evaluation in different indications. While the SYNERGY trial failed to demonstrate that the addition of an anti-CD73 targeting the adenosine immunosuppressive pathway to first-line chemotherapy with PD-L1 ICI does improve the CBR in unselected patients with mTNBC, it provides additional evidence that chemo-immunotherapy is highly effective at least in a subset of patients highlighting the heterogeneity of this disease and the urgent need to identify effective predictive biomarkers to tailor combination immunotherapy better.

## Methods

This research complies with all relevant ethical regulations. The SYNERGY trial was approved by the following ethical committees/authorities: Comité d'éthique hospitalo-facultaire Erasme-ULB and Federal Agency for medicines and health products (Belgium), and Comité de protection des personnes Ile de France and Agence nationale de sécurité du medicament et des produits de santé (France).

The study design and conduct complied with all relevant regulations regarding the use of human study participants and was conducted in accordance with the criteria set by the Declaration of Helsinki. All patients signed written informed consent before inclusion.

The SYNERGY trial was registered on ClinicalTrials.gov (https://clinicaltrials.gov/) (ID: NCT03616886).

## Study design

Phase I part examined the combination of 12 intravenous administrations of weekly paclitaxel 80 mg/m² and carboplatin AUC 2 in combination with 1500 mg durvalumab every four weeks with oleclumab every 2 weeks for five administrations and then every 4 weeks to define the RP2D of oleclumab in this combination. The first dose level of oleclumab was determined at 3000 mg according to a previous phase I study evaluating oleclumab in combination with durvalumab[36]. The DLT period evaluation was defined as the time from receiving the first dose of oleclumab until the planned administration of the third dose corresponding to 28 days period. The first patient was enrolled in the phase I on the 7 January 2019 and the sixth patient on the 8 March 2019.

In the phase II part, patients were randomly assigned (1:1), stratified by baseline PD-L1 and CD73 status, and by the site to one of the two arms. In arm A, patients received 12 intravenous administrations of weekly 80 mg/m² paclitaxel and AUC 1.5 of carboplatin in combination with 1500 mg durvalumab every 4 weeks with 3000 mg oleclumab every 2 weeks for five administrations and then every 4 weeks. Dosing regimens for oleclumab and durvalumab were based on phase I combination studies and were confirmed tolerable in combination with paclitaxel and carboplatin in the prior phase I part of SYNERGY[15,36]. In arm B, patients received the same chemotherapy regimen with durvalumab every 4 weeks without oleclumab. Durvalumab with or without oleclumab were administered intravenously until disease progression, withdrawal of consent, or unmanageable toxicity. The first patient was enrolled on the 19 June 2019 and the last patient on the 25 June 2021. Dose reductions for chemotherapy were allowed according to investigator practice but not for immunotherapy agents. Growth factor support could be added according to the prescribing information. The study design is available in Supplementary Fig. 1. Tumor and blood samples were collected for translation research purposes, including a metastatic lesion biopsy at baseline and on treatment during week 3.

## Patient population

Eligible patients were enrolled from 16 centers in Belgium and France. The complete list of inclusion and exclusion criteria is provided below.

Inclusion criteria:

1. Age of ≥18 years
2. Female
3. Life expectancy of a least 12 weeks
4. Body weight above 35 kg
5. The locally recurrent or metastatic relapse must be histologically confirmed TNBC in patients not previously treated with systemic treatment and which cannot be treated with curative intent. Newly diagnosed patients with de-novo metastatic disease are eligible
6. Estrogen receptor (ER) and progesterone receptor (PR) negativity (<1% positive staining cells in the invasive tumor) determined locally using IHC per ASCO/CAP criteria
7. Human epidermal growth factor receptor 2 (HER2) negativity (negative IHC staining [score 0 or 1] or negative fluorescence in situ hybridization [FISH] based on the ASCO/CAP guidelines and recommendations) and determined locally

   Note: patients initially diagnosed with hormone receptor–positive and/or HER2-positive breast cancer OR de novo metastatic patients with a primary tumor hormone receptor-positive (weak positivity or ER negativity and PR positivity) considered as non-clinically relevant are eligible if the tumor biopsy obtained from a local recurrence or distant metastasis site confirms the TNBC disease.
8. Confirmed tumor PD-L1 and CD73 IHC assessment as documented through central testing of a representative tumor tissue specimen for stratification purposes (only for phase II).
9. Provision of recurrence/metastatic tissue samples from resections, core-needle biopsies or excisional, incisional, punch, or forceps biopsies:

At least 1 FFPE [Formalin-Fixed paraffin-embedded] tumor tissue and 1 frozen core as a priority, if feasible 2 additional fresh tumor tissue cores should be collected too.

Fine-needle aspiration (FNA) (defined as samples that do not preserve tissue architecture and yield cell suspension and/or smears), brushing, and cell pellets from cytology samples are not acceptable.

Note 1: If the subject has just performed a metastatic lesion biopsy, the patient is eligible only if a FFPE tissue sample (or at least 20 unstained slides, freshly cut for the purposes of the study) of the metastatic/recurrent lesion is available. In this situation only, frozen cores are not mandatory.

Note 2: In case of a de-novo metastatic disease, if the biopsy of a metastatic lesion is not feasible, the patient is eligible if primary tumor lesion samples (FFPE + frozen cores) are available.

10. Provision of an archived FFPE diagnostic biopsy or surgical primary breast tumor sample (or at least 20 unstained slides, freshly cut for the purposes of the study).

Note: In case of neoadjuvant treatment (before surgery), the diagnostic biopsy is preferable.

11. At least 6 months elapsed between the completion of surgical and/or systemic treatment with curative intent (e.g., the date of primary breast tumor surgery or the date of last adjuvant chemotherapy administration (radiotherapy is not included), whichever occurred last) and first documented local or distant disease recurrence (NOTE: not applicable for de-novo metastatic disease)

12. At least one measurable disease based on RECIST v1.1. Tumor lesions in a previously irradiated area are considered measurable, if progression has been demonstrated in such lesions

13. Adequate organ function:
    a. Absolute neutrophil count (ANC) ≥ 1500/µl (without the addition of growth factors)
    b. Platelets [PLT] ≥100,000/µl (without the addition of growth factors/prior transfusions)
    c. Hemoglobin (Hb) ≥ 10 g/dl (without the addition of growth factors/prior transfusions)
    d. Creatinine ≤1.5 × upper limit of normal (ULN) OR estimated glomerular filtration rate (eGFR) ≥ 60 ml/min as calculated using the method standard for the institution. If eGFR is lower than 60 ml/min, a 24-h urine creatinine clearance can be performed to rule out an underestimation of the eGFR.
    e. Total serum bilirubin (TBL) ≤ 1.5 × ULN unless the subject has documented Gilbert syndrome in which case up to 3 x ULN is acceptable
    f. Aspartate and alanine aminotransferase (AST/ALT) ≤ 2.5 × ULN unless liver metastases are present, in which case it must be ≤5 × ULN
    g. International Normalized Ratio (INR) ≤ 1.5 × ULN unless subject is receiving anticoagulant therapy as long as INR and activated partial thromboplastin time (aPTT) is within therapeutic range of intended use of anticoagulants

14. Performance status (PS) of 0 or 1 on the ECOG Performance scale

15. Female subjects of childbearing potential (FSCP) must be willing to use one highly effective method of contraception for the course of the study through 6 months after the last study drug administration. FSCP must have a negative serum pregnancy test done within the 28 days before treatment start. FSCP are those who have not been surgically sterilized or have not been free of menses for at least 1 year.

16. Absence of any psychological, familial, sociological or geographical condition potentially hampering compliance with the study protocol and follow-up schedule; those conditions should be discussed with the patient before registration in the trial

17. Absence of any concurrent illness that would preclude the evaluation of safety

18. Agreement to provide tissue and blood samples for research purposes

19. Written informed consent must be given according to ICH/GCP, and national/local regulations before patient enrolment
    Inclusion criterion applicable to FRANCE only

20. Affiliated to the French Social Security System

Exclusion criteria:

1. Active or prior documented autoimmune or inflammatory disorders (including inflammatory bowel disease [e.g., colitis or Crohn's disease], diverticulitis [with the exception of diverticulosis], systemic lupus erythematosus, Sarcoidosis syndrome, or Wegener syndrome [granulomatosis with polyangiitis, Graves' disease, rheumatoid arthritis, hypophysitis, uveitis, etc]). The following are exceptions to this criterion:
    a. Patients with vitiligo or alopecia
    b. Patients with hypothyroidism (e.g., following Hashimoto syndrome) stable on hormone replacement
    c. Any chronic skin condition that does not require systemic therapy
    d. Patients without active disease in the last 5 years may be included but only after consultation with the sponsor
    e. Patients with celiac disease controlled by diet alone

2. Current or prior treatment with immunosuppressive medication within 14 days prior to enrolment. The following are exceptions to this criterion:
    a) Intranasal, inhaled, topical steroids, or local steroid injections (e.g., intra articular injection)
    b) Systemic corticosteroids at physiologic doses not to exceed 10 mg/day of prednisone or its equivalent
    c) Steroids as premedication for hypersensitivity reactions (e.g., CT scan premedication)

3. Any live, attenuated vaccine administered within 28 days prior to enrolment or anticipation that such a live attenuated vaccine will be required during the study

4. Chronic daily treatment with non-steroidal anti-inflammatory drug (NSAID) (occasional use for the symptomatic relief of medical conditions, for example, headache, fever is allowed)

5. Active infection including
    a. Tuberculosis (TB) (clinical evaluation that includes clinical history, physical examination and radiographic findings, and TB testing in line with local practice)
    b. Hepatitis B (known positive HBV surface antigen (HBsAg) result). Patients with a past or resolved HBV infection (defined as the presence of hepatitis B core antibody [anti-HBc] and absence of HBsAg) are eligible.
    c. Hepatitis C. Patients positive for hepatitis C (HCV) antibody are eligible only if polymerase chain reaction is negative for HCV RNA.
    d. Human immunodeficiency virus (positive HIV 1/2 antibodies).

6. Treatment with systemic immunostimulatory agents, including but not limited to, interferon (IFN)-alpha, IFN-beta, interleukin (IL) −2, conjugated IL-2 cytokines within 42 days or five half-lives of the drug, whichever is longer, prior to screening

7. Previous treatment with immune checkpoint inhibitors (e.g. anti-PD-1, anti-PD-L1 including durvalumab, anti-cytotoxic T-lymphocyte-associated molecule-4), anti-CD73 antibodies, adenosine A2A receptor antagonists, or prior treatment with CD137 agonists/OX-40 agonists or any other antibody or drug targeting T cell co-stimulation or other immunomodulatory therapies

8.  Any unresolved toxicity NCI CTCAE Grade ≥2 from previous anticancer therapy with the exception of alopecia, vitiligo and the laboratory values defined in the inclusion criteria

9.  Known hypersensitivity reactions to the study drugs or to any of the excipients, pre-medications (acetaminophen/paracetamol, diphenhydramine or equivalent anti-histamine and methylprednisolone or equivalent glucocorticoid) and to other platinum containing compounds

10. Untreated central nervous system (CNS) metastases and/or carcinomatous meningitis.

    Subjects with previously treated brain metastases with local treatment (stereotactic radiosurgery or whole brain radiation therapy) may participate provided they have stable brain metastases on a recent brain MRI (performed during the 2 weeks prior inclusion) and have measurable disease outside the CNS.

    Note: Known brain metastases are considered active (and not eligible for trial), if any of the following criteria are applicable:

    a.  Recent brain imaging demonstrates progression of existing and/or appearance of new lesions

    b.  Neurological symptoms attributed to brain metastases have not returned to baseline

    c.  Steroids were used for management of symptoms related to brain metastases within 14 days of enrolment

    d.  Completion of local therapy for brain metastases within 28 days of enrolment

11. Major surgical procedure (as defined by the principal investigator) within 28 days prior to enrolment. Note: Local surgery of isolated lesions for palliative intent is acceptable.

12. Uncontrolled intercurrent illness, including but not limited to,

    a.  Symptomatic congestive heart failure, uncontrolled hypertension, unstable angina pectoris, cardiac arrhythmia. Patients previously treated with anthracyclines are eligible if a recent cardiac work up (<6 months) demonstrated a normal left ventricular ejection fraction (LVEF ≥ 50%).

    b.  Interstitial lung disease

    c.  Serious chronic gastrointestinal conditions associated with diarrhea

    d.  Psychiatric illness/social situations that would limit compliance with study requirement, substantially increase risk of incurring AEs or compromise the ability of the patient to give written informed consent

13. Past medical conditions, including,

    a.  Class II-IV congestive heart failure

    b.  Myocardial infarction within 12 months prior enrolment,

    c.  Deep vein thrombosis (DVT) or thrombo-embolic event within 12 months prior to enrolment

    d.  History of stroke or transient ischemic attack requiring medical therapy

    e.  Intra-abdominal inflammatory process within the last 12 months prior to enrolment such as, but not limited to, diverticulitis, peptic ulcer disease, or colitis

    f.  History of idiopathic pulmonary fibrosis, organizing pneumonia (e.g. bronchiolitis obliterans), drug-induced pneumonitis, idiopathic pneumonitis, or evidence of active pneumonitis

    g.  History of another primary malignancy except for malignancy treated with curative intent and with no known active disease ≥5 years before the first dose of IP and of low potential risk for recurrence, adequately treated non-melanoma skin cancer or lentigo maligna without evidence of disease, adequately treated carcinoma in situ without evidence of disease

    h.  Status post allogeneic bone marrow transplantation or solid organ transplantation

14. Pregnant or lactating women.

15. Vulnerable persons according to the article L.1121-6 of the Public Health Code, adults who are the subjects of a measure of legal protection or unable to express their consent according to article L.1121-8 of the Public Health Code (Exclusion criterion applicable to FRANCE only).

## Study objectives and endpoints

Primary objective was to evaluate the clinical benefit (CB; complete response [CR] + partial response [PR] + stable disease [SD]) of adding the anti-CD73 antibody oleclumab to the combination of paclitaxel, carboplatin with durvalumab in previously untreated, locally recurrent inoperable or mTNBC patients by comparing the clinical benefit rate (CBR) after 24 weeks from the first dose of study drug administration between patients treated with or without oleclumab.

Secondary objectives were the following: (1) to compare the objective response rate (ORR; complete response [CR] + partial response [PR]) and the duration of response (DOR) between patients treated with or without oleclumab; (2) to compare PFS and overall survival (OS) between patients treated with or without oleclumab; (3) to evaluate the safety of the treatment combination; (4) to evaluate the efficacy of oleclumab in combination with paclitaxel, carboplatin and durvalumab according to PD-L1 and CD73 status.

Initial disease status was evaluated by imaging studies (contrast-enhanced CT scan and/or MRI of chest, abdomen and pelvis) during the screening phase. Disease status was followed by imaging studies at weeks 8 (±3 days), 16 (±3 days), and 24 (±3 days) after the start of treatment. Thereafter, imaging was performed every 12 weeks (±3 days; contrast-enhanced CT scan or MRI) regardless of any treatment delays. Patients who stopped all study treatments for reasons other than PD continued post-treatment imaging studies for efficacy follow-up until verified PD, start of a new anticancer treatment, withdrawal of consent to study participation, death, or end of the study (whichever came first).

Patients experiencing PD at week 8 or week 16, as defined by RECIST v1.1, could continue the study treatment in case of good clinical condition assessed by a stable or even improved ECOG performance status, to avoid treatment discontinuation in case of pseudo-progression. If the following assessment of tumor burden (8 weeks later) confirmed PD (as defined by RECIST), study treatment was discontinued.

The primary endpoint was the clinical benefit, defined as the achievement of CR or PR or demonstrated SD at 24 weeks from the first dose of study treatment based on the RECIST v1.1.

The secondary endpoints were the following: (1) Objective Response. OR was defined as achieving a CR or PR as the best overall response (BOR) based on RECIST v1.1; (2) Duration of Response. DOR is defined as the time from documentation of the first tumor response to disease progression based on RECIST v1.1; (3) Progression-Free Survival. PFS was defined as the time from the first study drug administration to the first documented disease progression based on RECIST v1.1 or death due to any cause, whichever occurs first. Patients who were alive and progression-free at the time of analysis were censored at the time-point of their last tumor assessment by imaging; (4) Overall Survival. OS was defined as the time from the first study drug administration to death due to any cause. Patients without documented death at the time of the analysis were censored at the date of the last follow-up; (5) Frequency, duration, and severity of AEs based on CTCAE 5.0.

## Statistical analyses

The primary analysis population was a modified intention-to-treat (mITT) population, which was defined as all randomized patients who were eligible and received at least one dose of treatment. These patients were considered evaluable for efficacy analysis. Patients were

analyzed in the arm they were allocated. A patient was considered to be eligible if they did not have any major deviations from the patient entry criteria. The assessment of the presence of major deviations was assessed centrally by the principal investigator and her medical team.

CBR, ORR, DOR, PFS, and OS were estimated in the mITT population.

The trial was powered to detect an increase in CBR from 40% to 60% at week 24. Therefore, a sample size of 136 (68 per arm) evaluable patients was required (calculated using EAST, with one-sided alpha = 0.1, power of 80%, and Casagrande–Pike–Smith correction).

An efficacy interim analysis for futility was performed after the enrolment of 68 evaluable patients (34 per treatment arm). The interim analysis was carried out when CBR assessment was possible for all patients, i.e., maximum 24 weeks for the last of the 68 evaluable patients. The primary purpose of efficacy interim analysis was to check the likelihood of observing the expected treatment effect. For planning the interim analysis, a Lan-Demets approach was used (O'Brien and Fleming spending functions for β error). On the $z$-scale, the pre-specified boundary for futility was 0.074, which implies the trial would be stopped for futility if the $z$-score is under this value. Safety and efficacy results were reviewed by an independent data monitoring committee (IDMC).

Comparisons of binary endpoints (Clinical Benefit, Objective Response) by arms or other categorical variables were performed using Fisher's exact tests. Non-parametric Wilcoxon rank-sum tests were used to compare continuous variables. Possible associations between two continuous variables were assessed using Spearman's correlation. Logistic regression models were used to assess the probability of the occurrence of binary events explained by some variables of interest and their interactions, represented by the odds ratios. Time-to-event outcomes, such as OS, PFS curves, the medians or survival at any time-point were estimated using Kaplan–Meier analysis. Log-rank test was used to compare the distributions of the survival curves. Cox regression analysis was used to estimate the hazard ratio between two survival curves and to assess the association between the time-to-event outcomes and potential prognostic factors. Univariate and multivariate models (logistic for binary and Cox for time-to-event outcomes) have been used to explore the associations between outcomes and these factors. Statistical tests were using one-sided for the primary efficacy analyses specified in the protocol and two-sided for exploratory subgroup analyses. If not stated in the text, the test is based on a two-sided test.

The safety population was defined as patients who received at least one dose of the assigned study treatment. AEs were graded by National Cancer Institute Common Terminology Criteria for Adverse Events (CTCAE) v5.0, and the safety data were descriptively compared between the arms.

Statistical analyses were done in SAS Enterprise Guide version 8.3. Graph Pad Prism version 9.5.1 was used for graphical representation.

## PD-L1 scoring
PD-L1 (VENTANA SP263 assay) IHC assessment was prospectively performed on a baseline tumor lesion on the VENTANA Benchmark at the Institut Jules Bordet (central laboratory). PD-L1 positivity threshold was set at ≥1% stained tumor and/or immune cells of any intensity relative to the tissue area (tumor and stroma). For exploratory biomarker analyses, baseline PD-L1 was retrospectively reviewed by a well-trained pathologist who was blinded for treatment group and clinical outcome. The PD-L1 staining was identified on immune versus tumor cells, and a combined positive score (CPS) defined as the number of positive tumor cells, lymphocytes, and macrophages divided by the total number of viable tumor cells multiplied by 100 was calculated[37]. In addition, PD-L1 was scored on week 3 biopsies to evaluate the changes in PD-L1 expression after treatment with chemo-immunotherapy.

## CD73 scoring
CD73 (Abcam EPR6115 clone, ab124725, 1:250 dilution) IHC assessment was performed on a baseline tumor lesion on the VENTANA Benchmark at the Institut Jules Bordet (central laboratory). CD73 positivity threshold was set at ≥1% of the stained tumor and/or stromal cells relative to the tissue area (tumor and stroma). For exploratory biomarker analyses, the CD73 staining was quantified as histological score (*H*-score), calculated as a percentage of tumor or stroma-stained cells multiplied by the intensity of staining in the tumoral or stromal compartments[35]. For each section, the percentages of stroma and tumoral compartments were based on morphology assessment.

## TIL assessments
TIL (intra-epithelial as well as stromal) levels were evaluated and quantified by trained pathologists using hematoxylin and eosin (H&E) stained tumor sections following the "The Immuno-Oncology Biomarker Working Group " guidelines (www.tilsinbreastcancer.org).

## Reporting summary
Further information on research design is available in the Nature Portfolio Reporting Summary linked to this article.

## Data availability
The raw and processed data generated in this study have been deposited at the Data Centre at Institut Jules Bordet in Brussels (Belgium) and can be made available upon approval of a research proposal. Any request for data (e.g., individual de-identified participant data, additional study documents including study protocol and/or statistical analysis plan) will be reviewed by the study team and should be addressed to L.B. at laurence.buisseret@bordet.be. Restrictions may apply to requests from industry or for commercial purposes. The expected timeframe for response to access requests is 6 months. Once access has been granted the data will be available for 12 months (extendible upon approval). Source data are provided with this paper.

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

## Acknowledgements

The authors thank all patients and their families. We thank the Institut Jules Bordet as Sponsor of the study. SYNERGY was designed by the authors in collaboration with the Sponsor. This study was conducted with support from AstraZeneca/MedImmune, which provided the study drugs (durvalumab, oleclumab) and a funding contribution for the trial. None of the funders had any role in the study design, data collection, data analysis, data interpretation, or writing of the report. All authors agreed and share the final responsibility for the provided interpretation of the study results and for the decision to submit for publication. We thank all medical staff from the participating sites. We thank CTSU from Institut Jules Bordet for every step of support in the development and conduct of the trial and the previous fellow from the Academic Trials Promoting Team, Mariana Brandao. We are grateful to the BCTL team of the Institut Jules Bordet for the central bio-banking of all biological samples and for helping in the conduct of translational analyses. Thanks, in particular to Delphine Vincent, Alfonsa Laragione, Mathilde Yernaux, Marion Maetens. Thanks to the Center for Cancer Immunotherapy, Institut Curie, Paris, France, and in particular Christina Metoikidou and the Laboratory for Translational Genetics, Department of Human Genetics, VIB Center for Cancer Biology, KU Leuven, Leuven, Belgium and in particular Diether Lambrechts for their collaboration in the translational research. The IDMC committee who reviewed in May 2021 the interim analysis.

We thank the MCCR Workshop Faculty for assistance in the protocol development during its attendance by Christian Maurer (2017 session). We are also grateful to the Association Jules Bordet and to the Fondation contre le Cancer for supporting the associated translational research projects.

## Author contributions

Study conception, design, funding, analysis, manuscript writing, study chair: L.B. Study conception, design, co-study chair: C.M. Study conception, translational research: J.S. Study conception and manuscript writing: M.I., M. Piccart. Manuscript writing: L.B., P.A., V.D., E.A., M.I., M. Piccart. Recruitment, clinical care and data return: D. Loirat, K.P., P.A. (Country principal investigator in Belgium), A.G., F.G., D.T., F.C. (Country principal investigator in France), T.V.D.M., J.-M.F., H.B., J.-L.C., F.D., L.M., R.P., P.B., N.I. Medical monitoring, data cleaning, data interpretation: V.B. (research fellow), E.A. (research fellow), D.E. (research fellow), E.d.A. Study pathologist: D. Larsimont. Biomarker assessments: L.C., Z.D., D. Larsimont, X.C., R.S. Data visualization: L.B., Z.D., D.V., V.D., P.K. Data analysis: P.K., M. Paesmans (study statisticians), L.B. Translational research: L.B., E.R., J.S., F.R., D.V., C.S. All authors approved the final version of this manuscript.

## Competing interests

None of the authors have conflicts of interest directly related to the present work. V.D., P.K., J-L.C., L.M., R.P., Z.D., X.C., R.S., F.R., L.C., D.V., D.L., M. Paesmaens: no competing interests. L.B.: financial interests, institutional, funding: AstraZeneca; financial interests, personal, funding: Fondation contre le Cancer. D.L.: financial interests, personal, advisory board, honoraria: AstraZeneca, Eli Lilly, Immunomedics, MSD, Pfizer, Roche, 4D Pharma, Daiichi Sankyo, Novartis, Gilead; financial interests, personal, royalties, travel support: Pfizer, Roche, MSD, AstraZeneca. P.A.: financial interests, personal, advisory board, consulting: Boehringer Ingelheim, Macrogenics, Roche, Novartis, Amcure, Servier, G1 Therapeutics, Radius, Deloitte; financial interests, personal, advisory board, honoraria: Synthon, Amgen, Novartis, Gilead; financial interests, personal, royalties, travel support: Amgen, MSD, Pfizer, Roche; financial interests, institutional, research grant: Roche. C.M.: financial interests, personal, royalties, travel support: Mundipharma, Amgen, Servier Deutschland GmbH, AbbVie; financial interests, personal, advisory board, honoraria: AbbVie; financial interests, personal, advisory board: Celgene/BMS. K.P.: financial interests, institutional, advisory board: AstraZeneca, Novartis, Roche, Vifor Pharma, Eli Lilly, Pierre Fabre, McCann Health, Roularta, Teva, Gilead Sciences, Pfizer, Gilead, MSD; financial interests, institutional, invited speaker: Pfizer, Roche, Novartis, Eli Lilly, Mundi Pharma, MSD, Medscape, MSD; financial interests, institutional, other, consultancy: Roche; financial interests, personal, other, consultancy: Gilead, Novartis, MSD, Roche; financial interests, personal, invited speaker: AstraZeneca, Sanofi; financial interests, personal, advisory board: Seagen; financial interests, institutional, funding: Sanofi; non-financial interests, principal investigator: EORTC 1745-ETF-BCG trial; non-financial interests, other, committee member: ESMO Young Oncologists Committee; non-financial interests, invited speaker: BSMO; non-financial interests, other, committee member: ESMO Resilience Task Force; non-financial interests, advisory role: Commission personalized medecine Federal Government Belgium; non-financial interests, advisory role, external scientific advisor: European Medicine Agency. D.E.: financial interest, employed, stocks/shares in F.Hoffman-La Roche Ltd. A.G.: research funding to the Institut Paoli-Calmettes from MSD, Bristol Myers Squibb, Novartis, Boerhinger Roche, Sanofi, Daiichi Sankyo, and AstraZeneca; personal fees to the Institut Paoli-Calmettes from MSD, Seagen, and Novartis. F.G: financial interests, personal, advisory board: AstraZeneca, Pierre Fabre, Roche, Amgen, Servier, Sanofi, MSD, BMS. D.T.: financial interests, personal, advisory board: AstraZeneca, Daiichi Sankyo, Eli Lilly, Medscape, Roche, Novartis, Agendia, MSD;

financial interests, personal, royalties: Pfizer, Roche, AstraZeneca; advisory board for Agendia, AstraZeneca, Daiichi Sankyo, Eli Lilly, Medscape, MSD, Novartis, Roche; travel fees from AstraZeneca, Pfizer, Roche. F.C.: financial interests, personal, advisory board: AstraZeneca, Eli Lilly, Roche, Merck, BMS, MSD; financial interests, institutional, research grant: AstraZeneca, Roche; financial interests, personal, royalties: AstraZeneca, Roche, Eli Lilly, Merck, BMS, MSD, Pfizer. T.v.d.M.: financial interests, personal, advisory board: Pfizer, Astellas, Bayer, MSD; financial interests, personal, royalties: Roche, Merck, Pfizer. J.-L.C.: financial interests, personal, advisory board: Lilly, Roche, Pfizer, BMS, Daiichi; financial interests, institutional, research grant: Roche, BMS, Amgen. J.M.F.: consulting or advisory roles for Roche, Pfizer, Novartis, and Eisai. H.B.: consulting or advisory role for AstraZeneca/Daiichi Sankyo; research funding from Bayer (Inst), travel, accommodations, expenses from Pfizer, AstraZeneca/Daiichi Sankyo. F.P.D.: financial interests, institutional, advisory board: Amgen, AstraZeneca, Daiichi Sankyo, Lilly, Gilead, Novartis, Pfizer, Pierre Fabre, Roche and Seagen, Amgen, AstraZeneca, Daiichi Sankyo, Lilly, Novartis, Pfizer, Pierre Fabre, Roche and Seagen, Gilead; Financial Interests, Personal, Royalties: Amgen, Pfizer, Roche, Teva. F. Bazan: financial interests, personal, advisory board: Roche, Pfizer, Novartis, AstraZeneca, Clovis, SeaGen, Daiichi Sankyo. P.B.: Amgen, Astellas Pharma, AstraZeneca, Bristol-Myers Squibb, Ipsen, Janssen-Cilag, Merck KGaA, MSD, MSD Oncology, Pfizer (C/A), Astellas Pharma, Bristol-Myers Squibb, Ipsen, Janssen-Cilag, MSD, Pfizer (travel expenses and accommodations); N.I.: consulting for Ipsen and Transgene; receipt of travel support from PharmaMar and Pfizer; and advisory board participation for Daiichi Sankyo. E.A.: consultancy fee/honoraria from Eli Lilly, Sandoz, AstraZeneca. Research grant to my institution from Gilead; support to attend medical conferences (travel/accommodation/expenses) from Novartis, Roche, Eli Lilly, Genetic, Istituto Gentili, Daiichi Sankyo, AstraZeneca (all outside the submitted work). E.d.A.: financial interests, personal, advisory board: Roche/GNE, Novartis; financial interests, personal, invited speaker: Seattle Genetic, Zodiac, Libbs, Pierre Fabre; financial interests, institutional, research grant: Roche/GNE, AstraZeneca, GSK/Novartis, Servier; financial interests, institutional, other, travel grant: Roche/GNE. E.R.: funding to her institution from Bristol-Myers Squibb (BMS), AstraZeneca, Janssen-Cilag, Replimmune, and from Fonds Amgen France pour la Science et l'Humain and travel support from BMS, Hoffmann La Roche, AstraZeneca, Merck Sharp & Dohme. J.S.: permanent member of the Scientific Advisory Board and owns stocks of Surface Oncology, is member of the Scientific Advisory Board of Tarus Therapeutics, and is a member of the Scientific Advisory Board of Domain Therapeutics. C.S.: advisory board (receipt of honoraria or consultations fees): Astellas, Cepheid, Vertex, Seattle genetics, Puma, Amgen, Exact Sciences; participation in company sponsored speaker's bureau: Eisai, Prime Oncology, Teva, Foundation Medicine, Exact Sciences; other support (travel, accommodation expenses): Roche, Genentech, Pfizer; M.I.: financial interests, personal, invited speaker: Novartis; financial interests, personal, advisory board: Novartis; financial interests, personal, other, independent monitoring committee: Seattle Genetics; financial interests, institutional, invited speaker: Pfizer, Roche, Natera; non-financial interests, invited speaker: EORTC; non-financial interests, officer: EORTC. M.P.: financial interests, personal, invited speaker: AstraZeneca, Lilly, MSD, Novartis, Pfizer; financial interests, personal, other, consultant: Camel-IDS/Precirix; financial interests, personal, advisory board: Immunomedics, Menarini, Odonate, Seattle Genetics, Immutep, SeaGen, Gilead, NBE Therapeutics, Frame Therapeutics; financial interests, personal, advisory board, consultant and invited speaker: Roche-Genentech; financial interests, personal, invited speaker, scientific board: Oncolytics; financial interests, institutional, research grant: AstraZeneca, Immunomedics, Lilly; financial interests, institutional, funding: Menarini, MSD, Novartis, Pfizer, Radius, Roche-Genentech, Servier, Synthon. The other authors declare no competing interests.

## Additional information

Laurence Buisseret [1] ✉, Delphine Loirat[2], Philippe Aftimos[1], Christian Maurer [3], Kevin Punie [4], Véronique Debien[1], Paulus Kristanto [1], Daniel Eiger[1], Anthony Goncalves[5], François Ghiringhelli [6], Donatienne Taylor [7], Florent Clatot[8], Tom Van den Mooter[9], Jean-Marc Ferrero[10], Hervé Bonnefoi[11], Jean-Luc Canon[12], Francois P. Duhoux [13], Laura Mansi[14], Renaud Poncin[15], Philippe Barthélémy[16], Nicolas Isambert[17], Zoë Denis[1], Xavier Catteau[18], Roberto Salgado [19], Elisa Agostinetto [1], Evandro de Azambuja[1], Françoise Rothé[1], Ligia Craciun[1], David Venet[1], Emanuela Romano [20], John Stagg [21], Marianne Paesmans[1], Denis Larsimont[1], Christos Sotiriou [1], Michail Ignatiadis [1] & Martine Piccart-Gebhart [1]

[1]Université Libre de Bruxelles (ULB), Hôpital Universitaire de Bruxelles (HUB), Institut Jules Bordet, 1070 Brussels, Belgium. [2]Medical Oncology Department, Institut Curie, 75005 Paris, France. [3]Department I of Internal Medicine, Center for Integrated Oncology Aachen Bonn Cologne Duesseldorf, University of Cologne, 52074 Cologne, Germany. [4]Department of General Medical Oncology and Multidisciplinary Breast Unit, Leuven Cancer Institute, University Hospitals Leuven, 3000 Leuven, Belgium. [5]Medical Oncology Department, Institut Paoli-Calmettes, 13274 Marseille, France. [6]Medical Oncology Department, Centre Georges-François Leclerc, 21000 Dijon, France. [7]Department of Oncology, CHU-UCL-Namur - Site Sainte-Elisabeth, 5000 Namur, Belgium. [8]Medical Oncology Department, Centre Henri Becquerel, 76038 Rouen, France. [9]Department of Oncology, GZA Ziekenhuizen Campus Sint-Augustinus, 2610 Antwerp, Belgium. [10]Department of Oncology, Centre Antoine Lacassagne, 06189 Nice, France. [11]Medical Oncology Department, Institut Bergonié, 33000 Bordeaux, France. [12]Department of Oncology-Hematology, Grand Hôpital de Charleroi - Site Notre Dame, 6000 Charleroi, Belgium. [13]Medical Oncology Department, Cliniques Universitaires Saint-Luc (UCLouvain), 1200 Brussels, Belgium. [14]Department of Oncology, CHU Besançon - Hôpital Jean Minjoz, 25030 Besancon, France. [15]Medical Oncology Department, Clinique Saint-Pierre, 1340 Ottignies-Louvain-la-Neuve, Belgium. [16]Medical Oncology Department, Institut de Cancérologie Strasbourg Europe (ICANS), 67000 Strasbourg, France. [17]Medical Oncology Department, CHU Poitiers, 86000 Poitiers, France. [18]CurePath Laboratory (CHU Tivoli, CHIREC), 6040 Jumet, Belgium. [19]Department of Pathology, GZA-ZNA Hospitals, 2610 Antwerp, Belgium. [20]Centre for Cancer Immunotherapy, Medical Oncology Department, INSERM U932, Institut Curie, PSL Research University, 75005 Paris, France. [21]Centre de Recherche du Centre Hospitalier de l'Université de Montréal, Faculté de Pharmacie et Institut du Cancer de Montréal, Montréal, QC 11290, Canada. ✉e-mail: laurence.buisseret@bordet.be

