## [Peer Review File · Nature Communications]

Paclitaxel Plus Carboplatin and Durvalumab With or Without
Oleclumab for women with Previously Untreated Locally
Advanced or Metastatic Triple-negative Breast Cancer: the
randomized SYNERGY phase I/II trialEditorial Note: This manuscript has been previously reviewed at another journal that is not operating a transparent peer review scheme. This document only contains reviewer comments and rebuttal letters for versions considered at *Nature Communications*.

REVIEWERS' COMMENTS

Reviewer #1 (Remarks to the Author):

Thank you for addressing the comments in your revision. I have a minor question about the p-value in Supplementary 1b, which has notably increased from 0.045 to 0.8194. Could you please provide clarification on the methods used for the previous and current analyses that could account for this change?

Reviewer #3 (Remarks to the Author):

After the revisions, the overall quality of the manuscript has significantly improved

Minor Consideration

1) Line 289-290 "A str-TIL levels increase of at least 5% on the week 3 biopsy was observed in 26% of the cases (6 cases in arm A (20%) and 9 cases in arm B (32.1%))." The authors should include a sentence clearly stating that there isn't a significant increase in str-TILS between baseline and week 3, referencing the results from the Wilcoxon paired test ($p=0.3969$). Additionally, the authors should report the percentage of patients who experienced a decrease of at least 5% by treatment arm.

Point by point responses to the Reviewer's comments

Manuscript: SYNERGY a randomized phase II trial of first-line chemo-immunotherapy with durvalumab, paclitaxel and carboplatin with or without the anti-CD73 antibody oleclumab in advanced triple-negative breast cancer (TNBC).

Reviewer #1 (Remarks to the Author):

Thank you for addressing the comments in your revision. I have a minor question about the p-value in Supplementary 1b, which has notably increased from 0.045 to 0.8194. Could you please provide clarification on the methods used for the previous and current analyses that could account for this change?

- *The two-sided log-rank test has been used to do the comparison between the two curves (Please find below the SAS output). The correct p-value is 0.814. There was an error in the previous version in the figure in manually copying the p-value from the SAS results to the GraphPad figure.*

Reviewer #3 (Remarks to the Author):

After the revisions, the overall quality of the manuscript has significantly improved.

- *Thank you very much for your positive feed-back and your help to improve the quality of our manuscript.*

Minor Consideration

1) Line 289-290 "A str-TIL levels increase of at least 5% on the week 3 biopsy was observed in 26% of the cases (6 cases in arm A (20%) and 9 cases in arm B (32.1%))." The authors should include a sentence clearly stating that there isn't a significant increase in str-TILS between baseline and week 3, referencing the results from the Wilcoxon paired test ($p=0.3969$). Additionally, the authors should report the percentage of patients who experienced a decrease of at least 5% by treatment arm.

- *A sentence to clearly state that this increase is not significant has been added in the manuscript.*
- *The percentage of patients who experienced a decrease of at least 5% is now mentioned in the manuscript in addition of the supplementary table.*